# Sage extract and ascorbic acid derivative inhibit melanogenesis via downregulating keratinocyte-derived GM-CSF

Hirokazu Kubo[1]°, Mariko Moriyama[2]°, Saya Goto[2], Yuko Miyake[2], Maki Nakamura[1], Yuki Ozeki[1], Yukio Nakamura[1], Hiroyuki Moriyama[2]*

**1** R&D Headquarters, KOBAYASHI Pharmaceutical Co., Ltd., Ibaraki, Osaka, Japan, **2** Pharmaceutical Research and Technology Institute, Kindai University, Higashi-Osaka, Osaka, Japan

☙ These authors contributed equally to this work.
* moriyama@phar.kindai.ac.jp

## Abstract

*Salvia officinalis* (sage) extract has demonstrated potential as a functional ingredient for skin care application. However, its effect and mechanism in regulating skin pigmentation remain largely unclear. This study investigated the effects of sage ethanol extract (SGE) on melanogenesis and its underlying molecular mechanisms. Treatment with SGE in a human skin equivalent model (3D-skin) suppressed melanin production. To clarify the mechanism of action, the study focused on senescence-associated secretory phenotype (SASP) factors, which are implicated in age-related pigmentation changes. q-PCR and ELISA analyses showed that SGE inhibits melanogenesis by suppressing the expression of granulocyte-macrophage colony-stimulating factor (GM-CSF), a known SASP factor in keratinocytes. Interestingly, a similar effect was observed with L-ascorbic acid 2-glucoside (AG), previously identified as a tyrosinase inhibitor. Importantly, p38 and JNK MAP-kinase were identified as upstream regulators of GM-CSF that are suppressed by SGE. These findings provide new insights into how SGE and AG regulate pigmentation via keratinocyte-derived GM-CSF, highlighting their potential in modulating skin tone and pigmentation through cellular signaling pathways.

## Introduction

Melanin is synthesized in melanosomes, organelles found in melanocytes, through the action of tyrosinase. Once synthesized, melanin-containing melanosomes are secreted from the dendritic tips of melanocytes and transferred to surrounding epidermal cells [1]. In these epidermal cells, melanosomes are localized above the nucleus to protect the nuclear DNA from ultraviolet (UV) radiation-induced damage [2]. The accumulation of melanosomes in epidermal cells is usually transient and reversible, as seen in phenomena like a suntan.

**Data availability statement:** All relevant data are within the manuscript and its Supporting information files.

**Funding:** The author(s) received no specific funding for this work.

**Competing interests:** The authors have declared that no competing interests exist.

Melanogenesis is tightly controlled by intra-melanocyte and extra-melanocyte signaling mechanisms. Notably, senescence-associated secretory phenotype (SASP) factors, secreted by senescent keratinocytes following UV exposure, play a significant role in this process. SASP factors, including granulocyte/macrophage colony-stimulating factor (GM-CSF), interleukin (IL)-1α, IL-6, transforming growth factor (TGF)-β, and transforming necrosis factor (TNF) α [3–11], create a pro-inflammatory environment that stimulates melanogenesis. Chronic exposure to UV radiation can further exacerbate this effect by inducing keratinocyte senescence, disrupting the balance of signaling pathways in the skin microenvironment. This disruption is linked to the development of solar lentigo, which manifests as pigment spots on the skin surface and is an important aesthetic concern associated with aging.

To mitigate excessive pigmentation, research has focused on identifying active compounds that suppress melanogenesis and melanosome removal. Commonly used agents include ascorbic acid (AA; vitamin C), nicotinamide (vitamin B3), and hydroquinone. Of these, AA has long been recognized for its skin-brightening properties due to its high inhibitory effect on tyrosinase activity. However, AA is unstable when incorporated into cosmetic formulations. To address this, L-ascorbic acid 2-glucoside (AG), a stabilized form of AA created by adding glucose, has been developed. AG improves the stability of AA, making it more suitable for practical applications. In addition to enhancing stability, AG is also known to prevent UV-induced skin damage and reduce melanin production by inhibiting tyrosinase activity, similar to AA [12]. To further augment AG's brightening and anti-melanin effects, it is often combined with natural extracts that possess anti-aging properties in cosmetic formulations. Based on this evidence, we investigated various natural extracts to identify a complementary agent that could act synergistically with AG in suppressing melanin production.

The genus *Salvia* comprises about 1,000 species widely used in cosmetics, foods, and traditional medicines. These plants grow globally, with *Salvia officinalis* L. (sage) originating in the Middle East and Mediterranean regions. Sage contains various bioactive compounds, including alkaloids, carbohydrate, fatty acids, glycosidic derivatives, phenolic compounds, poly acetylenes, steroids, and terpenes/terpenoids [13,14]. Studies have shown that methanol extracts of sage exhibit anti-wrinkle properties by suppressing hyaluronidase and elastase [15], and they also demonstrate anti-melanin effects by suppressing tyrosinase activity at levels comparable to kojic acid, a well-known skin-brightening agent [16]. Furthermore, sage extract has been investigated for its ability to regulate melanin production, making it more suitable for practical applications. Though these findings underscore sage's potential as an anti-melanin agent, it is essential to recognize that the bioactive components of plant extracts can vary significantly depending on the extraction method. In particular, the brightening effects and specific anti-melanin mechanisms of ethanol extracts, which are commonly used in cosmetic formulations, have not been fully elucidated. Therefore, further investigation is required to clarify the potential of sage ethanol extracts in inhibiting melanin production and promoting skin brightening.

This study focused on the effects of SGE and AG on pigmentation. SGE treatment enhanced the brightening effect in 3D-skin models, although it did not directly

suppress melanin production in melanocytes. To investigate the mechanism behind this, the role of cytokines in melanin regulation was examined, and it was found that both SGE and AG suppress melanin production via GM-CSF. Specifically, SGE and AG attenuate UVB-induced GM-CSF expression in keratinocytes, which in turn influences melanocytes and reduces melanin synthesis. This regulation occurs through the downregulation of the p38 and JNK MAP-kinase pathways. These results suggest that SGE and AG play a crucial role in inhibiting melanogenesis by modulating keratinocyte-derived cytokines. This is particularly significant given the widespread use of sage-based products in medical and cosmetic applications.

## Materials and methods

### Test substances

SGE was obtained by extraction with ethanol solution from the leaf of sage (*Salvia officinalis* L.). In brief, ethanol solution was added to the leaves of *Salvia officinalis* L. (Labiatae), followed by extraction, standing in a cold place, purification, filtration, and freeze-drying to obtain the final product. It was purchased from ICHIMARU PHARCOS Co., Ltd. (Gifu, Japan). Details of the bioactive ingredients of SGE used in these experiments are shown in Table 1. AG was purchased from HAYASHIBARA Co., Ltd. (Okayama, Japan). Other extracts using screening are shown in S1 Table.

### Reagents

Recombinant human GM-CSF (Prospec, CYT-221, Rehovot, Israel) was used at a concentration of 20 pg/mL to stimulate cell activity. The neutralizing antibody (LSBio, LS-C104671-50, rabbit polyclonal antibody, Seattle, WA, USA) was used at a concentration of 0.2 µg/mL to inhibit GM-CSF. SB203580 (Selleck, Houston, TX, USA) was used at a concentration of 10 µM to inhibit the p38 MAPK pathway during treatment. SP600125 (Selleck) was applied at a concentration of 10 µM to block JNK signaling.

### Cell culture

3D-skin (MEL-300-A) was purchased from MatTek Corporation (Ashland, MA, USA) and maintained in EPI-100LLMM (MatTek Corporation) culture medium according to the manufacturer's instructions. Human Primary Epidermal Keratinocytes (HPEKs) were purchased from CELLnTEC (Bern, Switzerland) and maintained in CnT-PR (CELLnTEC) culture medium according to the manufacturer's protocol. Normal Human Epidermal Melanocytes (NHEMs) were purchased from KURABO (Osaka, Japan) and maintained in DermaLife M Comp kit (KURABO) culture medium according to the manufacturer's protocol.

### UVB irradiation

Prior to UVB irradiation, the cells and the cultured skin were washed with PBS and irradiated with a handheld UV Lamp (UVP, Upland, CA, USA). The accuracy of the UVB dose was calculated using a UVX-31 radiometer (UVP).

**Table 1. Bioactive ingredients in SGE.**

| Ingredient | SGE (mg/g) |
| --- | --- |
| Homoplantaginin | 13.689 ± 0.0004 |
| Rosmarinic acid | 5.277 ± 0.002 |
| Scutellarin | 2.938 ± 0.001 |
| Luteolin | 2.367 ± 0.002 |
| Oleanolic acid | 0.047 ± 0.00006 |
| Ursolic acid | 0.0065 ± 0.0003 |

## Evaluation of melanin production in the 3D-skin model

The 3D skin models containing melanocytes were treated with 0.1 μg/mL extracts (shown in S1 Table) and/ or 2.5 mM AG for 11 days. UVB irradiation was applied every 2 days during the treatment period. 3D-skin was lysed with M-PER (Thermo Fisher Scientific, Waltham, MA, USA) following sonication. The lysate was centrifuged at 15,000 rpm for 15 min and separated into supernatant (protein) and melanin. The amount of protein in the supernatant was measured with the BCA Protein Assay Kit (Takara Bio Inc., Japan). After solubilizing melanin in 100 μL of 4N NaOH, the melanin content was measured using an absorbance meter (MULTISKAN FC, Thermo Fisher Scientific) at 405 nm. The melanin content was normalized to protein levels.

## Co-culture of NHEMs with HPEKs

NHEMs seeded onto a 6-well culture plate (Corning, Glendale, AZ, USA) at a density of $3 \times 10^5$ cells per well, while HPEKs were seeded on Falcon cell culture inserts (Permeable Support for 6-well plate with a 3.0-μm transparent PET membrane, Corning) at a density of $5 \times 10^5$ cells per insert. Following a 24-hour culture period, HPEKs were treated with either AG or SGE and incubated for 30 min, after which UVB irradiation at a dose of 5 mJ/cm$^2$ was administered. Co-culturing with NHEMs then commenced in CnT-PR medium.

## Measurement of melanin content of NHEMs

The harvested cells were washed twice with a mixture of ethanol and ether (1:1, v/v) at room temperature for 10 minutes. After discarding the organic solvent, the dried pellets were solubilized in 1 M sodium hydroxide containing 10% DMSO by incubation at 80 ˚C for 15 minutes. The absorbance of dissolved melanin was measured at 490 nm using a microplate reader (Revvity, Waltham, MA, USA). The concentration of protein in the cell lysate was determined using a QuickStart Bradford protein assay system (Bio-Rad Laboratories, Hercules, CA, USA). The ratio of the A490 to protein content was used for comparing intracellular melanin contents.

## RNA extraction, cDNA generation, and quantitative polymerase chain reaction

Total RNA was extracted using the RNeasy Mini Kit (Qiagen, Hilden, Germany) according to the manufacturer's instructions. The cDNA was generated from 1 μg of total RNA using the Verso cDNA Synthesis Kit (Thermo Scientific) and purified using the MinElute PCR Purification Kit (Qiagen). Quantitative polymerase chain reaction (q-PCR) analysis was conducted using the SsoFast EvaGreen supermix (Bio-Rad, Hercules, CA, USA) according to the manufacturer's protocols. The relative expression value for each gene was calculated using the ΔΔCt method, and the most reliable internal control gene was determined of eight genes (*ACTB*, *B2M*, *GAPDH*, *GUS*, *HPRD*, *RN18S*, *UBE2D2*, and *UBE4A*) using geNorm Software (Biogazelle, Zwijnaarde, Belgium). Details of the primers used in these experiments are shown in Table 2.

## ELISA assay

The levels of human Granulocyte-Macrophage Colony-Stimulating Factor (GM-CSF) in the cell culture supernatants of HPEKs were measured using the ELISA MAXStandard Set Human GM-CSF (BIOLEGEND, San Diego, CA, USA) according to the manufacturer's instructions. Briefly, ELISA plates were coated with anti-GM-CSF antibody, blocked, and then incubated with standards and HPEK supernatant samples. After washing, plates were incubated with biotinylated anti-GM-CSF antibody followed by HRP-conjugated streptavidin. Color development was initiated with TMB substrate, stopped, and absorbance was measured at 450 nm. GM-CSF concentrations were determined using a standard curve. All experiments were performed in triplicate, and the mean absorbance values were used for data analysis. Statistical analyses were carried out using GraphPad Prism software (version X; GraphPad Software Inc., San Diego, CA, USA).

**Table 2. Primers used in this study.**

| Name | Forward primer (5'-3') | Reverse primer (5'-3') |
|---|---|---|
| ACTB | CATGTACGTTGCTATCCAGGC | CTCCTTAATGTCACGCACGAT |
| B2M | TATCCAGCGTACTCCAAAGA | GACAAGTCTGAATGCTCCAC |
| GAPDH | CATGAGAAGTATGACAACAGCCT | AGTCCTTCCACGATACCAAAGT |
| GUS | CACCAGGGACCATCCAATACC | GGTTACTGCCCTTGACAGAGA |
| HPRT | ATTGTAATGACCAGTCAACAGGG | GCATTGTTTTGCCAGTGTCAA |
| RN18S | ATCCATTGGAGGGCAAGTC | GCTCCCAAGATCCAACTACG |
| UBE2D2 | TGGCAAGCTACAATAATGGGG | GGAGACCACTGTGATCGTAGA |
| UBE4A | GTACTTGGGATTTCACAGGTTGC | GGCTAGAACTTTGCTGAGCATC |
| CSF2 | TCCTGAACCTGAGTAGAGACAC | TGCTGCTTGTAGTGGCTGG |

## Western blot analysis

Cells were lysed with lysis buffer (20 mM Tris-HCl (pH 8.0), 1% SDS, and 1 mM DTT). Blots were probed with a rabbit monoclonal antibody against SAPK/JNK (Clone 56G8; Cell Signaling Technology; #9258), rabbit monoclonal antibody against phospho SAPK/JNK (T183/Y185; Clone 81E11; Cell Signaling Technology; #4668), rabbit polyclonal antibody against p38 (Cell Signaling Technology; #9212), rabbit monoclonal antibody against phospho p38 (T180/Y182; Clone 3D7; Cell Signaling Technology; #9215), and mouse monoclonal antibody against actin (Clone C4; MAB1501; Merck-Millipore, Billerica, MA, USA). Horseradish peroxidase (HRP)-conjugated anti-mouse or rabbit IgG secondary antibody (Cell Signaling Technology; #7076, #7074) was used as a probe, and immunoreactive bands were visualized with the Immobilon Western Chemiluminescent HRP substrate (Merck-Millipore). Band intensity was measured using ImageJ version 1.49 (NIH, Bethesda, MD, USA).

## Identification of bioactive ingredients in SGE

SGE was reconstituted in 50% ethanol to a concentration of 5 µg/L after freeze-drying. A standard solution for constructing a calibration curve, as described previously, was used, and compounds present in the sage extract were identified and quantified using the standard solution dissolved in methanol. Quantitative analysis of the standard solution and sage sample was carried out using LC/MS. The HPLC system used was the LC-20AD (Shimadzu, Kyoto, Japan), and the LCMS used was the LCMS-8040 (Shimadzu). The column used was the COSMOSIL 5C18-AR-II, 5 um, 4.6 mm I.D. × 150 mm (Nacalai Tesque, Kyoto, Japan). The mobile phase comprises 10 mM ammonium acetate in water (solvent A) and acetonitrile (solvent B). The gradient was as follows: 0 min, 5% B; 5 min, 5% B; 15 min, 20% B; 20 min, 25% B; 30 min, 30% B; 40 min, 40% B; 45 min, 50% B; 50 min, 80% B; 53 min, 80%B; 55 min, 5%B; 60 min, 5% B. The injection volume for the sage sample was 20 µL, and for the standard solution, it was 5 µL. All analyses were performed at 35 °C. Each component was identified based on its retention time in chromatography and its MS spectrum. Quantification was performed by comparing the integration peak area with the calibration curve made using corresponding analytical standards. Isomeric compounds such as ursolic acid and oleanolic acid were differentiated based on their retention times and negative ionization mode fragment mass-to-charge ratio (m/z) values. All HPLC analyses were performed in triplicate.

## Statistical analysis

Statistical analyses were performed using GraphPad Prism 10. Depending on the experimental design, one-way ANOVA followed by Dunnett's or Tukey's test, or two-way ANOVA followed by Šídák's test was used. Data are expressed as mean ± standard error values. The value of $P < 0.05$ was considered significant.

## Results

### Enhancement of the skin-brightening effect of AG by SGE in a 3D-skin model

To identify natural extracts that additively inhibit melanin production with AG, screening was conducted using a 3D-skin model. Approximately 80 natural extracts that can be utilized in Japanese quasi-pharmaceutical cosmetic products were tested in a preliminary screening for this study. As a result, sage ethanol extract (SGE) was identified as a candidate because it showed the most significant inhibition of melanin production among the tested extracts (S1 Table). Although treatment with SGE alone did not reach statistical significance, it showed an additive effect in reducing melanin levels in the 3D-skin model when combined with AG (Fig 1). These results demonstrate that SGE enhances the effect of AG in reducing melanin levels in the 3D-skin model.

### Indirect suppression of melanogenesis by SGE and AG

Previous studies have suggested that sage methanol extracts inhibit tyrosinase activity [16], leading us to hypothesize that SGE might similarly reduce melanin production in normal human epidermal melanocytes (NHEMs). Therefore, melanin levels were initially examined in NHEMs treated with SGE to explore its potential molecular mechanism. AG, already known to reduce melanin levels in melanocytes, was included as a positive control and was found to significantly inhibit melanin production as expected. In contrast, SGE did not affect melanogenesis in NHEMs (Fig 2a). These results suggest that SGE acts on keratinocytes and modulates melanin levels via cytokines released by surrounding keratinocytes.

To verify whether SGE's inhibitory effect on melanin production is related to cytokine regulation, an indirect co-culture system between melanocytes and keratinocytes was established. For this purpose, human primary epidermal keratinocytes (HPEKs) were seeded on inserts, treated with SGE, and exposed to UV irradiation. The inserts were then transferred to a culture plate containing melanocytes. After a two-day co-culture period, melanin levels in melanocytes were measured, showing a significant decrease in melanin production due to SGE and AG (Fig 2b).

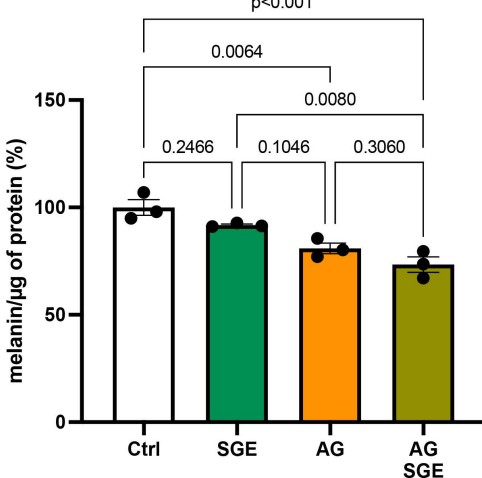

**Fig 1. SGE and AG additively reduce melanin production.** Human full-thickness skin equivalents containing melanocytes were treated with 0.1 µg/mL SGE and/or 2.5 mM AG for 11 days. UVB irradiation was applied every 2 days during the treatment period. Afterward, melanin content in the skin equivalents was measured and normalized to protein content. Data are presented as mean ± SEM values from three independent experiments. Statistical significance was calculated using one-way ANOVA followed by Dunnett's test.

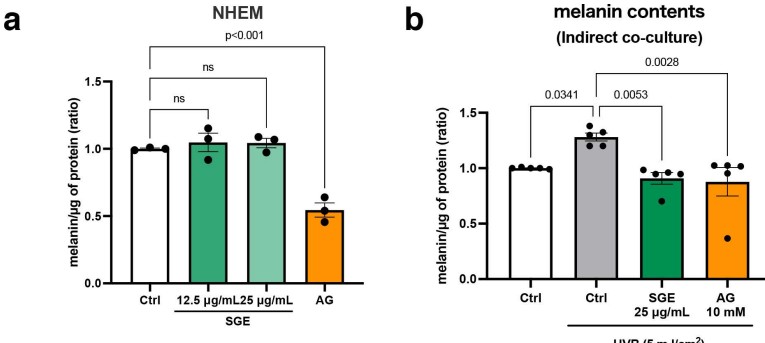

Fig 2. SGE inhibits melanogenesis indirectly through its effects on keratinocytes, rather than acting directly on melanocytes. (a) NHEMs were treated with SGE or 2.5 mM AG for 3 days, after which melanin content was measured and normalized to protein content. Data are presented as mean±SEM values from four independent experiments. (b) HPEKs were treated with SGE and AG, incubated for 30 min, and then exposed to UVB irradiation at a dose of 5 mJ/cm². Co-culturing with NHEMs was subsequently initiated in CnT-PR medium. Melanin contained in NHEMs was measured and normalized to protein content. Data are presented as mean±SEM values from five independent experiments. Statistical significance was calculated using one-way ANOVA followed by Dunnett's test.

## Molecular mechanism of inhibiting melanogenesis by SGE and AG

Next, the effects of SGE on SASP factors were evaluated, focusing on cytokines involved in melanogenesis. Of the tested mRNA expressions encoding cytokines, *CSF2*, which encodes GM-CSF, and *TGFB2*, which encodes TGF-β2, were significantly upregulated by UVB exposure. In contrast, SGE treatment following UVB irradiation led to a reduction in *CSF2* levels (Fig 3a and S1 Fig). In addition, ELISA analysis demonstrated that, though UV irradiation increased the levels of GM-CSF protein, SGE treatment led to a decrease in its levels (Fig 3b). To confirm whether GM-CSF is directly involved in melanin production, indirect co-culture experiments were conducted. As shown in Fig 2b, SGE treatment suppressed the elevated melanin levels induced by UVB irradiation, whereas the addition of GM-CSF canceled the anti-melanogenic effects of SGE, leading to increased melanin levels in NHEMs. In addition, when a neutralizing antibody against GM-CSF was added to the HPEKs treated with both SGE and GM-CSF, melanin levels in NHEMs decreased (Fig 3c). Interestingly, the data showed an unexpected finding that AG also attenuated the UVB-induced GM-CSF expression (Fig 3a, b). These data indicate that SGE and AG inhibit melanogenesis by negatively regulating keratinocyte-derived GM-CSF, providing a new perspective on AG beyond its already established antioxidant abilities and tyrosinase activity inhibition.

It has been reported that MAPK signaling is involved in the regulation of GM-CSF expression, with the promoter region of the *CSF2* gene containing a binding site for AP1, a downstream factor of p38-JNK MAPK signaling [17,18]. Our previous report also demonstrated that UVB activates MAPK signaling in keratinocytes [19]. Therefore, whether SGE contributes to the activation of MAPK signaling following UVB irradiation was investigated using western blotting. Activation of p38 and JNK MAPK were observed 4–8 hours after UVB irradiation, but their activation was significantly suppressed by the addition of SGE (Fig 4a). To determine whether the enhanced GM-CSF expression induced by UVB irradiation is also mediated by MAPK signaling, HPEKs were treated with specific MAPK inhibitors, SP600125 (JNK inhibitor) and SB203580 (p38 inhibitor), followed by UVB irradiation. The results indicated that *CSF2* expression induced by UVB irradiation was suppressed by the addition of both the JNK inhibitor and the p38 inhibitor (Fig 4b), confirming that SGE inhibits the expression of *CSF2* via JNK and p38 MAPK.

## Discussion

SGE has been used as a folk medicine because it has several important therapeutic properties, including anti-inflammation and wound healing. It also has applications in the cosmetic industry. However, the specific effects of SGE

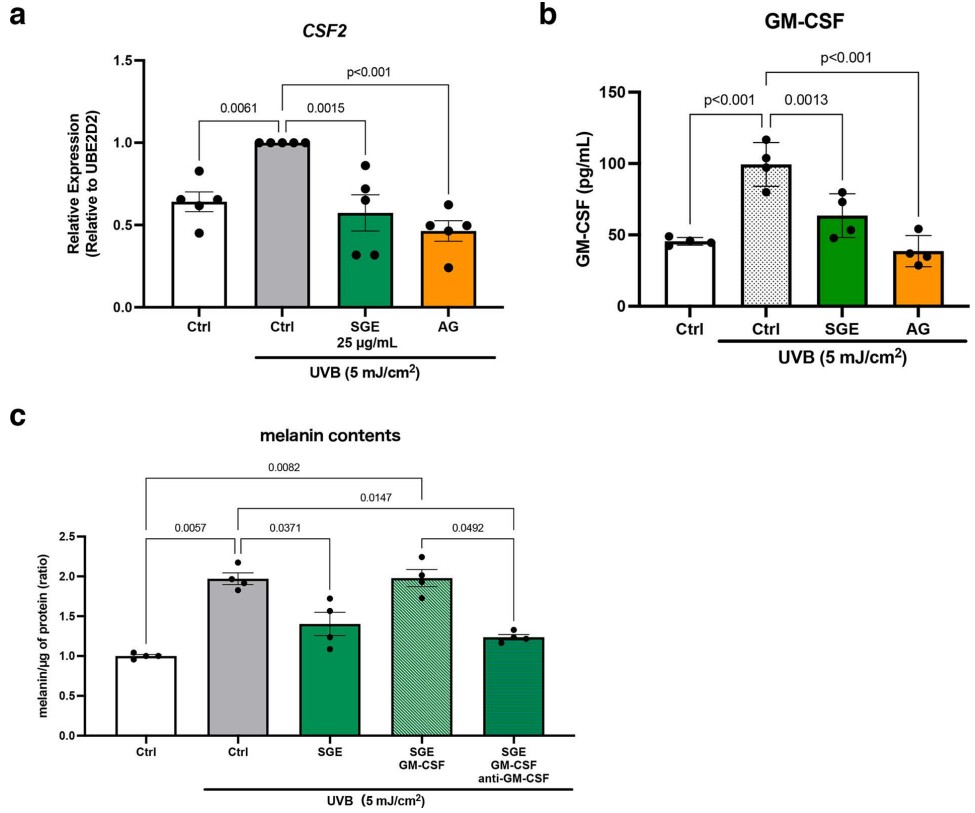

**Fig 3. SGE suppresses melanogenesis by inhibiting GM-CSF production in keratinocytes.** HPEKs were treated with SGE, AG, GM-CSF, and anti-GM-CSF antibody, incubated for 30 min, and then exposed to UVB irradiation at a dose of 5 mJ/cm². Subsequently, co-culturing with NHEMs was initiated in CnT-PR medium. Forty-eight hours after starting the co-culture, the following experiments were performed. (a) Quantitative PCR (q-PCR) analysis of *CSF2* expression in HPEKs. Data are presented as mean±SEM values from five independent experiments. (b) ELISA analysis of GM-CSF released from HPEKs. Data are presented as mean±SEM values from four independent experiments. (c) Measurement of melanin content in HPEKs, normalized to protein content. Data are represented as mean±SEM values from four independent experiments. Statistical significance was calculated using one-way ANOVA followed by Dunnett's test for (a) and (b), and Tukey's test for (c).

on epidermal keratinocytes and melanocytes, particularly in response to UV exposure, remain unexplored. In the present study, molecular and cellular approaches were used to generate scientific data on the effects of SGE on pigmentation, and active mechanisms were identified.

The skin, being the outermost layer of the body, is frequently exposed to UV radiation. UV-exposed epidermal keratinocytes are known to secrete SASP factors, which influence the surrounding environment by altering intercellular communication and contributing to various skin conditions. Of them, solar lentigo is a notable example, characterized by hyperpigmentation caused by chronic UV exposure and SASP factor secretion [20–22]. In general, hyperpigmentation disorders require treatment with anti-pigmenting agents. Most of these agents have been developed as tyrosinase inhibitors, whereas others target melanogenesis signaling pathways. However, the majority of such treatments focus on intracellular mechanisms within melanocytes [23,24]. In contrast, the results of the present study highlight the importance of targeting the extracellular environment that influences melanocytes, a relatively unexplored approach in the development of anti-pigmenting agents. In this context, our screening approach, which emphasized the environment surrounding melanocytes, provides a novel perspective for the development of anti-pigmenting agents. The current study demonstrated, for the first time, that SGE and AG indirectly inhibit melanin production through the suppression of GM-CSF secretion derived from keratinocytes, rather than directly targeting

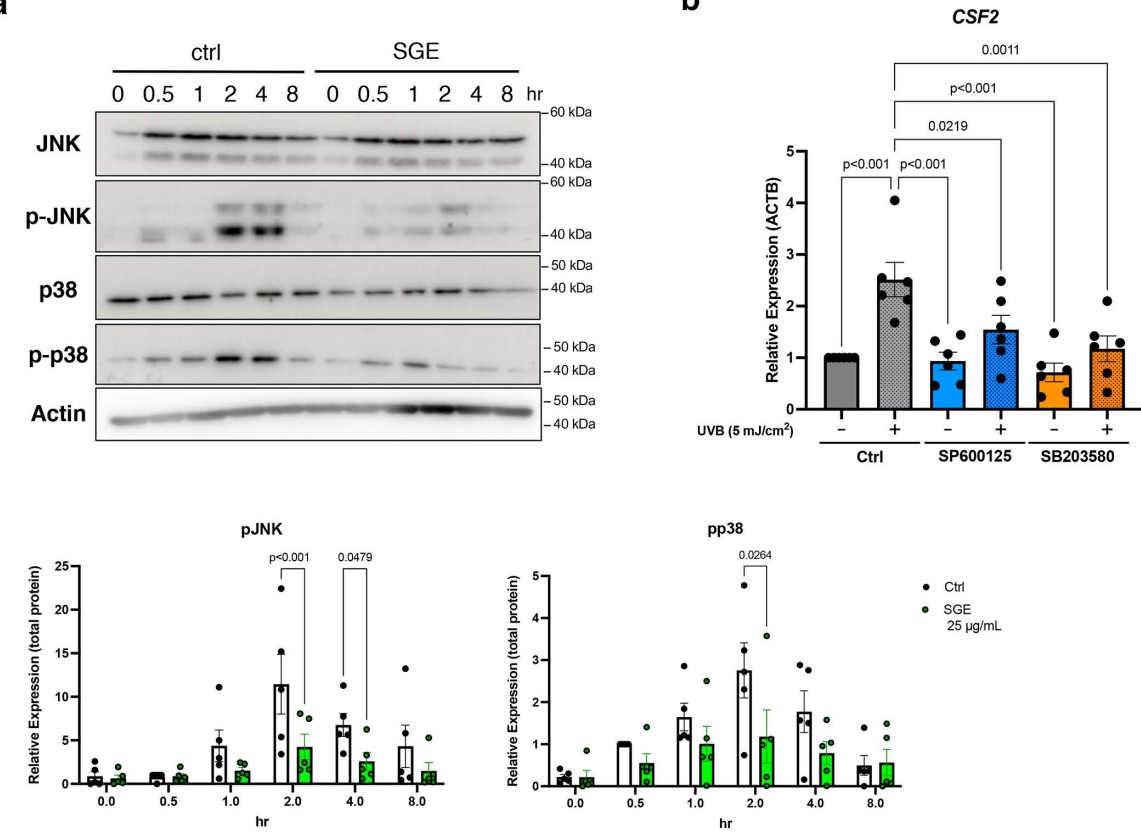

**Fig 4. *CSF2* expression is regulated by p38 and JNK MAPK in response to UVB exposure.** (a) HPEKs were exposed to UVB radiation (5 mJ/cm$^2$) and incubated for the indicated times. The extracted proteins were then immunoblotted with the indicated antibodies. Graphs show the relative band intensities as determined by ImageJ and plotted as the mean ± SEM values of five independent experiments. (b) q-PCR analysis of *CSF2* expression in HPEKs. HPEKs were exposed to UVB radiation (5 mJ/cm$^2$) and incubated for 72 h. SP600125 (10 µM) and SB203580 (10 µM) were added 1 h before UVB exposure. Data are presented as mean ± SEM values from six independent experiments. Statistical significance was calculated using two-way ANOVA followed by Šídák's test (a) or one-way ANOVA followed by Dunnett's test (b).

melanocytes (Figs 2 and 3). In addition, the experiments using a 3D-skin model showed that both AG and SGE reduce melanin production and melanosome accumulation (Fig 1 and S1 Table). These effects appear to stem from not only the direct inhibition of tyrosinase activity, but also the suppression of SASP factor secretion. This suggests that regulating SASP factor secretion could represent a promising strategy for controlling hyperpigmentation disorders. Intriguingly, AG has also been shown to inhibit TGF-β1, TGF-β2, and FGF7 (Fig 3 and S1 Fig).

Given that GM-CSF is a critical SASP factor involved in pigmentation, targeting keratinocyte-derived GM-CSF may offer dual benefits for addressing pigmentation and age-related skin changes. Screening strategies focusing on SASP factors secreted from keratinocytes hold the potential to not only advance the development of anti-pigmenting agents, but also contribute to the creation of anti-aging agents for skin. This is because inhibiting SASP factors can reduce inflammation, improve tissue homeostasis, and alleviate age-related changes in skin structure and function by mitigating the harmful effects of senescent cells [22,25]. This approach could establish a new avenue in cosmetic science. Furthermore, SASP factors are secreted not only by keratinocytes, but also by other skin-resident cells, including senescent fibroblasts [25,26]. Given that fibroblasts play a critical role in maintaining skin architecture and extracellular matrix composition,

SASP factors derived from these cells may also influence melanogenesis. Future studies focusing on SASP factors derived from fibroblasts will likely become increasingly important.

SASP factor expression is regulated by various intracellular signaling pathways, among which JNK and p38 MAPK play pivotal roles in promoting SASP-related transcription. Activation of these pathways is well-established to enhance transcription factors such as NF-κB and AP-1, driving the expression of pro-inflammatory cytokines, including IL-6 and GM-CSF [17,18,27]. This mechanism is particularly relevant under stress conditions such as UV damage, where JNK and p38 MAPK pathways are prominently activated in keratinocytes [19]. The present study targeted these pathways to investigate their roles in UVB-induced GM-CSF expression. Notably, it was demonstrated that SGE suppresses UVB-activated JNK and p38 MAPK, thereby reducing the expression of *CSF2* (GM-CSF), which is primarily mediated through these pathways (Fig 4). These findings underscore the broader utility of targeting stress-induced pathways in keratinocytes for modulating SASP factors and their downstream effects of skin pigmentation and inflammation. Interestingly, both SGE and AG are known to exhibit anti-inflammatory effects through their antioxidant properties [28,29]. These anti-inflammatory properties have been widely recognized in the context of cosmetic and skincare applications. However, the present study is, to our knowledge, the first to link the anti-inflammatory effects of SGE and AG with their ability to modulate melanogenesis. The present findings highlight the suppression of GM-CSF, a keratinocyte-derived SASP factor, as a key mechanism underlying this effect. Future studies should aim to elucidate the broader molecular networks involved in these interactions. For instance, comprehensive gene expression profiling or proteomic analyses could provide deeper insights into how SGE and AG influence melanogenesis through intercellular communication. By focusing on the crosstalk between keratinocytes and melanocytes, we anticipate that these investigations will further uncover the therapeutic potential of SGE and AG as innovative agents for skin pigmentation disorders.

SGE contains a variety of bioactive compounds [13,14], including luteorin, scutellarin, homoplantaginin, and rosmarinic acid as identified by MS/MS analysis (Table 1). To further investigate the potential contribution of these compounds to the observed effects of SGE, their impact on GM-CSF expression was examined individually. Though none of the tested compounds at concentrations equivalent to those in the SGE formulation significantly reduced GM-CSF expression, higher doses of rosmarinic acid (30-fold) and luteolin (2-fold) did exhibit significant suppression (Supplemental S2 Fig). These findings suggest that the suppressive effects of SGE on GM-CSF may result from the combined or synergistic action of its components, rather than the effect of a single compound. Notably, both luteolin and rosmarinic acid have been reported to exert anti-inflammatory effects by modulating oxidative stress and inflammatory signaling pathways. For example, luteolin is known to inhibit NF-κB and MAPK pathways, reducing the expression of pro-inflammatory cytokines such as IL-6 and IL-1β [30,31]. Similarly, rosmarinic acid has demonstrated antioxidant properties that mitigate ROS-induced activation of these pathways [32]. Though these mechanisms align with the present findings, the concentrations of these compounds required to achieve similar effects in isolation suggest that SGE's efficacy is enhanced by its complex composition.

Interestingly, the effects of sage extract on melanogenesis reported in the present study differ from previously published findings. Previous studies have reported that the sage extract and its compound have tyrosinase inhibition activity [16,33]. In contrast, Lee et al. demonstrated that rosmarinic acid induces melanogenesis in melanoma cells [34]. These discrepancies may be attributed to differences in experimental design, including the extraction methods used, the concentration of compounds tested, and the cellular models used. In the present study, ethanol-extracted sage extract was used, whereas previous research often used extracts derived using methanol or other solvents. The choice of solvent is known to significantly affect the composition and concentration of bioactive compounds within extracts. Both ethanol and methanol are effective solvents for extracting bioactive compounds, but their extraction profiles differ due to their chemical properties. Ethanol efficiently extracts both polar and non-polar compounds, including phenolic derivatives, lipids, and terpenoids, which are important for bioactivity. It is also considered safer for food and pharmaceutical applications. Methanol, in contrast, is particularly efficient in extracting highly polar compounds such as certain phenolic and polysaccharides. However, its toxicity limits its application in consumable products [35]. These solvent-specific properties likely contribute

to the differing compositions and activities of sage extracts obtained using these solvents. Moreover, the concentrations of rosmarinic acid reported to promote melanogenesis in melanoma cells (50 µM) are approximately 150 times higher than those present in the SGE used in the present study [34]. This raises concerns about the physiological relevance of those findings, since the concentrations used in the present study are closer to levels observed under normal conditions. In addition, the use of melanoma cells, which possess altered signaling pathways and metabolic profiles compared with normal melanocytes, further limits the direct comparability of the findings. In contrast, the UVB-induced keratinocyte environment used in this study may better represent the physiological context of pigmentation disorders. These distinctions emphasize the importance of considering both the experimental context and the cellular microenvironment when interpreting the effects of bioactive compounds.

In summary, this study provides the first demonstration that SGE and AG have inhibitory effects on melanogenesis via negatively regulating keratinocyte-derived GM-CSF. These should be further evaluated for their relevance to medicinal and cosmetic applications. This novel mechanism offers a foundation for developing multifunctional agents targeting pigmentation disorders and skin aging.

## Supporting information

**S1 Fig. Influence of SGE and AG on the expression of melanogenesis factors in human epidermal keratinocytes.** HPEKs were treated with either SGE (25 µg/mL) or AG (10 mM), incubated for 30 min, and then exposed to UVB irradiation at a dose of 5 mJ/cm2. Subsequently, co-culturing with NHEMs was initiated in CnT-PR medium. Forty-eight hours after starting the co-culture, the following experiments were performed. Gene expression levels of *IL1A*, *IL1B*, *IL6*, *FGF2*, *FGF7*, *TGFB1*, and *TGFB2* were quantified by q-PCR analysis. Details of the primers used in these experiments are shown in S2 Table. The graphs indicate the mean ± SEM values for relative expression from eight independent experiments. Statistical significance was calculated using one-way ANOVA followed by Dunnett's test.
(TIF)

**S2 Fig. The suppressive effects of SGE's components on GM-CSF.** q-PCR analysis of *CSF2* expression in HPEKs. HPEKs were exposed to UVB radiation (5 mJ/cm2) and incubated for 72 h. (a) HPEKs were treated with either SGE (25 µg/mL) or rosmarinic acid (0.125, 0.36, 1.2, 3.6 µg/mL) 30 min before UVB exposure. (b) HPEKs were treated with either SGE or scutellarin (0.25, 0.75, 2.5, 7.5 µg/mL) 30 min before UVB exposure. (c) HPEKs were treated with SGE, luteorin (0.06, 0.12 µg/mL), or homoplantaginin (0.25, 0.50 µg/mL) 30 min before UVB exposure. The graphs indicate the mean ± SEM values for relative expression from four to eight independent experiments. Statistical significance was calculated using one-way ANOVA followed by Dunnett's test.
(TIF)

**S1 Table. Sage ethanol extract (SGE) was identified as a candidate for anti-melanogenesis ingredient.** Approximately 80 natural extracts that can be utilized in quasi-pharmaceutical cosmetic products were tested in a preliminary screening for this study. These extracts were obtained safe extraction solution (ex. water, ethanol, 1,3-butylene glycol) for skin followed by freeze drying. The 3D-skin model was cultured for 11 days. UV-B irradiation, medium replacement, and the addition of PBS containing the natural extracts (0.1 µg/mL) were performed every other day. Melanin contents were measured following 11 days culture.
(PDF)

**S2 Table. Primer pairs used in this study.**
(PDF)

**S1_raw_images. Western blot raw data.** Original images of protein Western blot experiments used in the main figures.
(PDF)

## Acknowledgments

The authors would like to thank Miho Yorozu, Risa Furuta, Saki Inamura, Mayu Kadono, Fuka Shimoji, and Yuki Tamura for their technical support. The authors thank FORTE Science Communications (https://www.forte-science.co.jp/) for English language editing.

## Author contributions

**Conceptualization:** Hirokazu Kubo, Mariko Moriyama, Hiroyuki Moriyama.

**Data curation:** Hirokazu Kubo, Mariko Moriyama, Saya Goto, Yuko Miyake, Maki Nakamura, Yuki Ozeki, Hiroyuki Moriyama.

**Formal analysis:** Hirokazu Kubo, Mariko Moriyama.

**Funding acquisition:** Yukio Nakamura.

**Investigation:** Hirokazu Kubo, Mariko Moriyama, Saya Goto, Yuko Miyake, Maki Nakamura, Yuki Ozeki.

**Methodology:** Hirokazu Kubo, Mariko Moriyama, Hiroyuki Moriyama.

**Project administration:** Mariko Moriyama, Hiroyuki Moriyama.

**Supervision:** Mariko Moriyama, Hiroyuki Moriyama.

**Validation:** Hirokazu Kubo, Mariko Moriyama, Hiroyuki Moriyama.

**Visualization:** Hirokazu Kubo, Mariko Moriyama, Hiroyuki Moriyama.

**Writing – original draft:** Hirokazu Kubo, Mariko Moriyama, Hiroyuki Moriyama.

**Writing – review & editing:** Hirokazu Kubo, Mariko Moriyama, Hiroyuki Moriyama.

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
