## [Decision Letter · Decision Letter 0]

29 Mar 2025

Dear Dr. Moriyama,

Thank you for submitting your manuscript to PLOS ONE. After careful consideration, we feel that it has merit but does not fully meet PLOS ONE’s publication criteria as it currently stands. Therefore, we invite you to submit a revised version of the manuscript that addresses the points raised during the review process.

We look forward to receiving your revised manuscript.

Kind regards,

Anju George

Academic Editor

PLOS ONE

Journal Requirements:

Additional Editor Comments :

Dear authors,

Please find the comments of the reviewers below.

Reviewer 1:

Comments:

1. In page20 under results section, it is mentioned that many plant extracts (around 80) were tested before the decision was made to choose sage extract, what are the other substances tested and how did you arrive at the conclusion that sage extract works best with ascorbic acid? This process is not mentioned in the methodology and the figure 1 cited in the text only shows SGE and AG effects on melanin concentration, but not the other compounds.

2. In figure 3, the expression of CSF2 is compared to that of UBE2D2. What is the relevance of UBE2D2 expression in this model? This part has not been discussed anywhere. Only the use of that primer is mention and why was this chosen?

3. The discussion part mentions about variation in results due to ethanol versus methanol extraction leading to discrepancies in in previous studies. In this study ethanol was used as a solvent only for sage extract, however, for the control tissue and AG the solvent used was methanol as per the description. Why was the same agent not used and could it have influenced the results of your study?

Reviewer2:

Skin hyperpigmentation could lead to varied skin conditions, including pigment spots on skin surfaces; thus, there is the search for ingredients to be used in skin care products to minimise skin pigmentation and ageing. The authors showed that sage extract (SGE) and L-ascorbic acid 2-glucoside (ascorbic acid derivative; AG) inhibit melanogenesis through downregulating keratinocyte-derived GM-CSF, which is vital in making skincare and anti-ageing products. The manuscript should be accepted for publication with these minor revisions:

1. In the abstract, the authors introduced AG with no prior definition. To help readers, the authors should define all acronyms or abbreviations in their first use in the manuscript.

2. SGE was obtained by extraction with ethanol solution from the leaf of sage (Salvia officinalis L.) The authors should briefly describe how this extraction was done.

3. HPEKs were purchased from CELLnTEC (Bern, Switzerland), and NHEMs were purchased from KURABO. What are HPEKs and NHEMs? The authors should make it easier for readers to follow their experiments. A brief description of HPEKs and NHEMs should be included in the methods.

4. Statistical differences were evaluated using one-way analysis of variance (ANOVA), followed by Tukey’s test, Dunnett’s test, or Bonferroni’s test. Why the three different post hoc tests? Tukey’s test and Bonferroni’s test were not mentioned again in the results.

5. The authors stated that 80 plant extracts were evaluated in their study (line 223). However, nothing was said about these plants, their names, or their mode of extraction in terms of materials and methods.

6. The authors were looking for a synergistic effect between SGE and AG. However, there was no significant effect in the 3D-skin model (figure 1). I see this as an additive effect, not a synergistic effect because there was no fold decrease in the melanin levels. Their results demonstrate that SGE enhances the impact of AG in reducing melanin levels in the 3D-skin model. Enhancement could be synergistic or additive; therefore, the authors should clarify whether their report is additive or synergistic.

7. In the discussion of the manuscript, “SGE has been used as a folk medicine because it has several important therapeutic properties, including inflammation and wound healing” (lines 314-315). It should be revised to “SGE has been used as a folk medicine because it has several important therapeutic properties, including anti-inflammation and wound healing.”

Reviewers' comments:

Reviewer's Responses to Questions

**Comments to the Author**

1. Is the manuscript technically sound, and do the data support the conclusions?

Reviewer #1: Yes

Reviewer #2: Yes

2. Has the statistical analysis been performed appropriately and rigorously?

Reviewer #1: Yes

Reviewer #2: Yes

3. Have the authors made all data underlying the findings in their manuscript fully available?

Reviewer #1: No

Reviewer #2: Yes

4. Is the manuscript presented in an intelligible fashion and written in standard English?

Reviewer #1: Yes

Reviewer #2: Yes

Reviewer #1: Comments:

1. In page20 under results section, it is mentioned that many plant extracts (around 80) were tested before the decision was made to choose sage extract, what are the other substances tested and how did you arrive at the conclusion that sage extract works best with ascorbic acid? This process is not mentioned in the methodology and the figure 1 cited in the text only shows SGE and AG effects on melanin concentration, but not the other compounds.

2. In figure 3, the expression of CSF2 is compared to that of UBE2D2. What is the relevance of UBE2D2 expression in this model? This part has not been discussed anywhere. Only the use of that primer is mention and why was this chosen?

3. The discussion part mentions about variation in results due to ethanol versus methanol extraction leading to discrepancies in in previous studies. In this study ethanol was used as a solvent only for sage extract, however, for the control tissue and AG the solvent used was methanol as per the description. Why was the same agent not used and could it have influenced the results of your study?

Reviewer #2: Skin hyperpigmentation could lead to varied skin conditions, including pigment spots on skin surfaces; thus, there is the search for ingredients to be used in skin care products to minimise skin pigmentation and ageing. The authors showed that sage extract (SGE) and L-ascorbic acid 2-glucoside (ascorbic acid derivative; AG) inhibit melanogenesis through downregulating keratinocyte-derived GM-CSF, which is vital in making skincare and anti-ageing products. The manuscript should be accepted for publication with these minor revisions:

1. In the abstract, the authors introduced AG with no prior definition. To help readers, the authors should define all acronyms or abbreviations in their first use in the manuscript.

2. SGE was obtained by extraction with ethanol solution from the leaf of sage (Salvia officinalis L.) The authors should briefly describe how this extraction was done.

3. HPEKs were purchased from CELLnTEC (Bern, Switzerland), and NHEMs were purchased from KURABO. What are HPEKs and NHEMs? The authors should make it easier for readers to follow their experiments. A brief description of HPEKs and NHEMs should be included in the methods.

4. Statistical differences were evaluated using one-way analysis of variance (ANOVA), followed by Tukey’s test, Dunnett’s test, or Bonferroni’s test. Why the three different post hoc tests? Tukey’s test and Bonferroni’s test were not mentioned again in the results.

5. The authors stated that 80 plant extracts were evaluated in their study (line 223). However, nothing was said about these plants, their names, or their mode of extraction in terms of materials and methods.

6. The authors were looking for a synergistic effect between SGE and AG. However, there was no significant effect in the 3D-skin model (figure 1). I see this as an additive effect, not a synergistic effect because there was no fold decrease in the melanin levels. Their results demonstrate that SGE enhances the impact of AG in reducing melanin levels in the 3D-skin model. Enhancement could be synergistic or additive; therefore, the authors should clarify whether their report is additive or synergistic.

7. In the discussion of the manuscript, “SGE has been used as a folk medicine because it has several important therapeutic properties, including inflammation and wound healing” (lines 314-315). It should be revised to “SGE has been used as a folk medicine because it has several important therapeutic properties, including anti-inflammation and wound healing.”

**Do you want your identity to be public for this peer review?** For information about this choice, including consent withdrawal, please see our Privacy Policy

Reviewer #1: No

Reviewer #2: No

---

## [Author Response · Author response to Decision Letter 1]

20 Apr 2025

Response to Reviewer 1:

We wish to express our appreciation to the reviewers for their insightful comments on our paper. The comments have helped us significantly improve the paper.

1. In page20 under results section, it is mentioned that many plant extracts (around 80) were tested before the decision was made to choose sage extract, what are the other substances tested and how did you arrive at the conclusion that sage extract works best with ascorbic acid? This process is not mentioned in the methodology and the figure 1 cited in the text only shows SGE and AG effects on melanin concentration, but not the other compounds.

Response: We thank the reviewer for raising this important point. As part of the preliminary screening for this study, we tested approximately 80 commercially available natural extracts to evaluate their effects on melanogenesis using a 3D skin model containing melanocytes. These extracts, listed in the newly added S1 Table, were selected based on their suitability for use in Japanese quasi-pharmaceutical cosmetic products. All extracts were prepared using skin-safe solvents (e.g., water, ethanol, or 1,3-butylene glycol), then freeze-dried and reconstituted for testing.

The screening was performed according to the protocol now described in the revised Methods section (p. 5, lines 124–131). This screening revealed that sage ethanol extract (SGE) exerted the most significant inhibitory effect on melanin production among the tested extracts. Although treatment with SGE alone did not reach statistical significance in some cases, its combination with AG showed an additive effect, as presented in Figure 1.

To address the reviewer’s concern, we have revised both the Methods and Results sections accordingly.

Specifically:

Methods section (p. 5, lines 124–125):

The 3D skin models containing melanocytes were treated with 0.1 μg/mL extracts (shown in S1 Table) and / or 2.5 mM AG for 11 days. UVB irradiation was applied every 2 days during the treatment period. 3D-skin was lysed with M-PER (Thermo Fisher Scientific, Waltham, MA, USA) following sonication. The lysate was centrifuged at 15,000 rpm for 15 min and separated into supernatant (protein) and melanin. The amount of protein in the supernatant was measured with the BCA Protein Assay Kit (Takara Bio Inc., Japan). After solubilizing melanin in 100 μL of 4N NaOH, the melanin content was measured using an absorbance meter (MULTISKAN FC, Thermo Fisher Scientific) at 405 nm. The melanin content was normalized to protein levels.

Results section (p. 9, line 218 - 224):

To identify natural extracts that additively inhibit melanin production with AG, screening was conducted using a 3D-skin model. Approximately 80 natural extracts that can be utilized in Japanese quasi-pharmaceutical cosmetic products were tested in a preliminary screening for this study. As a result, sage ethanol extract (SGE) was identified as a candidate because it showed the most significant inhibition of melanin production among the tested extracts (S1 Table). Although treatment with SGE alone did not reach statistical significance, it showed an additive effect in reducing melanin levels in the 3D-skin model when combined with AG (Fig 1).

2. In figure 3, the expression of CSF2 is compared to that of UBE2D2. What is the relevance of UBE2D2 expression in this model? This part has not been discussed anywhere. Only the use of that primer is mention and why was this chosen?

Response: Thank you for your comment regarding the selection of the internal control gene in Figure 3. To identify the most stable reference gene for our qPCR analysis, we evaluated eight commonly used housekeeping genes (ACTB, B2M, GAPDH, GUS, HPRD, RN18S, UBE2D2, and UBE4A) using the geNorm software (Biogazelle, Zwijnaarde, Belgium). Among these candidates, UBE2D2 was determined to be the most stable gene under our experimental conditions, and was therefore used as the internal control.

We have added the primer information for these genes in Table 2, and included the following description in the Methods section　(p. 6, line 158-159):

The relative expression value for each gene was calculated using the ΔΔCt method, and the most reliable internal control gene was determined of eight genes (ACTB, B2M, GAPDH, GUS, HPRD, RN18S, UBE2D2, and UBE4A) using geNorm Software (Biogazelle, Zwijnaarde, Belgium).

3. The discussion part mentions about variation in results due to ethanol versus methanol extraction leading to discrepancies in in previous studies. In this study ethanol was used as a solvent only for sage extract, however, for the control tissue and AG the solvent used was methanol as per the description. Why was the same agent not used and could it have influenced the results of your study?

Response: We thank the reviewer for this thoughtful comment. As noted, discrepancies in previous studies may be attributed to differences in extraction solvents, such as methanol versus ethanol. We would like to clarify that in our study, methanol was not used at any step. The sage extract (SGE) was prepared using ethanol as the extraction solvent, then freeze-dried into a powdered form. For treatment, this powder was directly dissolved or suspended in culture medium.

AG, in contrast, is a pure compound and not a plant extract. It was provided as a powder and similarly dissolved directly in culture medium without the use of methanol. Therefore, no methanol was used for either SGE, AG, or control samples, and solvent differences are unlikely to have influenced the results of this study.

Thank you again for your comments on our paper. We trust that the revised manuscript is suitable for publication.

Response to Reviewer2:

We wish to express our strong appreciation to the reviewers for their insightful comments on our paper. We feel the comments have helped us significantly improve the paper.

1. In the abstract, the authors introduced AG with no prior definition. To help readers, the authors should define all acronyms or abbreviations in their first use in the manuscript.

Response: Thank you for this suggestion. We have revised the Abstract to define the abbreviation at first mention. The sentence now reads as follows (p. 2, line 29):

“Interestingly, a similar effect was observed with L-ascorbic acid 2-glucoside (AG), previously identified as a tyrosinase inhibitor.”

2. SGE was obtained by extraction with ethanol solution from the leaf of sage (Salvia officinalis L.) The authors should briefly describe how this extraction was done.

Response: Thank you for this suggestion. We have added the following description to the Methods section (p. 4, lines 93–95):

“In brief, ethanol solution was added to the leaves of Salvia officinalis L. (Labiatae), followed by extraction, standing in a cold place, purification, filtration, and freeze-drying to obtain the final product.”

3. HPEKs were purchased from CELLnTEC (Bern, Switzerland), and NHEMs were purchased from KURABO. What are HPEKs and NHEMs? The authors should make it easier for readers to follow their experiments. A brief description of HPEKs and NHEMs should be included in the methods.

Response: Thank you for this suggestion. In accordance with the reviewer's comment, we have revised the Methods section (p. 5, lines 112–116) to include the following description:

“Human Primary Epidermal Keratinocytes (HPEKs) were purchased from CELLnTEC (Bern, Switzerland) and maintained in CnT-PR (CELLnTEC) culture medium according to the manufacturer’s protocol. Normal Human Epidermal Melanocytes (NHEMs) were purchased from KURABO (Osaka, Japan) and maintained in DermaLife M Comp kit (KURABO) culture medium according to the manufacturer’s protocol.”

4. Statistical differences were evaluated using one-way analysis of variance (ANOVA), followed by Tukey’s test, Dunnett’s test, or Bonferroni’s test. Why the three different post hoc tests? Tukey’s test and Bonferroni’s test were not mentioned again in the results.

Response: We apologize for the confusion and thank the reviewer for their careful observation. Statistical significance in this study was evaluated using only Dunnett’s test. Tukey’s test and Bonferroni’s test were not applied.

We have corrected the description of the statistical analysis in the Methods section (p. 9, lines 208–209) as follows:

“Statistical differences were evaluated using one-way analysis of variance (ANOVA), followed by Dunnett’s test using GraphPad Prism 10.”

In addition, we have revised the figure legends for Figures 1–4 to accurately reflect the statistical method used.

5. The authors stated that 80 plant extracts were evaluated in their study (line 223). However, nothing was said about these plants, their names, or their mode of extraction in terms of materials and methods.

Response: Thank you for pointing this out. In response to similar comments, we have now included the names and details of the approximately 80 natural extracts used in the screening as a new supplementary table (S1 Table). These plant extracts are commercially available and are permitted for use in Japanese quasi-pharmaceutical cosmetic products. Each extract was obtained using skin-safe solvents such as water, ethanol, or 1,3-butylene glycol, followed by freeze-drying.

We have also updated the Materials and Methods section (p. 5, lines 124–131) to clarify how these extracts were used in our screening experiment.

We hope this revision improves the clarity and reproducibility of our methodology.

6. The authors were looking for a synergistic effect between SGE and AG. However, there was no significant effect in the 3D-skin model (figure 1). I see this as an additive effect, not a synergistic effect because there was no fold decrease in the melanin levels. Their results demonstrate that SGE enhances the impact of AG in reducing melanin levels in the 3D-skin model. Enhancement could be synergistic or additive; therefore, the authors should clarify whether their report is additive or synergistic.

Response: Thank you for bringing this to our attention. We agree with your interpretation and have revised the manuscript accordingly. In the Results section (p. 9, lines 222–224), we have changed the wording to:

“Although treatment with SGE alone did not reach statistical significance, it showed an additive effect in reducing melanin levels in the 3D-skin model when combined with AG (Fig 1).”

7. In the discussion of the manuscript, “SGE has been used as a folk medicine because it has several important therapeutic properties, including inflammation and wound healing” (lines 314-315). It should be revised to “SGE has been used as a folk medicine because it has several important therapeutic properties, including anti-inflammation and wound healing.”

Thank you for your suggestion. We agree and have revised the sentence in the Discussion section (p. 12, line 314) to:

“SGE has been used as a folk medicine because it has several important therapeutic properties, including anti-inflammation and wound healing.”

Thank you again for your comments on our paper. We trust that the revised manuscript is suitable for publication.

---

## [Decision Letter · Decision Letter 1]

5 May 2025

Dear Dr. Moriyama,

Thank you for submitting your manuscript to PLOS ONE. After careful consideration, we feel that it has merit but does not fully meet PLOS ONE’s publication criteria as it currently stands. Therefore, we invite you to submit a revised version of the manuscript that addresses the points raised during the review process.

We look forward to receiving your revised manuscript.

Kind regards,

Anju George

Academic Editor

PLOS ONE

Journal Requirements:

Additional Editor Comments:

Thanks to the authors for revising their manuscript per the comments raised by the reviewers. The manuscript should be accepted for publication with these minor revisions:

1. The authors agreed that what is reported is additive rather than synergistic. However, the title of Figure 1 still maintains synergistic. I will suggest that all synergistic effects in the manuscript be changed to additive effects since there was no fold decrease in the melanin levels.

2. In the materials and methods section, “Other extracts using screening were shown in S1 Table” should be revised to “Other extracts used in the screening are shown in S1 Table.”

Reviewers' comments:

Reviewer's Responses to Questions

**Comments to the Author**

Reviewer #2: All comments have been addressed

2. Is the manuscript technically sound, and do the data support the conclusions?

Reviewer #2: Yes

3. Has the statistical analysis been performed appropriately and rigorously?

Reviewer #2: Yes

4. Have the authors made all data underlying the findings in their manuscript fully available?

Reviewer #2: Yes

5. Is the manuscript presented in an intelligible fashion and written in standard English?

Reviewer #2: No

Reviewer #2: Thanks to the authors for revising their manuscript per the comments raised by the reviewers. The manuscript should be accepted for publication with these minor revisions:

1. The authors agreed that what is reported is additive rather than synergistic. However, the title of Figure 1 still maintains synergistic. I will suggest that all synergistic effects in the manuscript be changed to additive effects since there was no fold decrease in the melanin levels.

2. In the materials and methods section, “Other extracts using screening were shown in S1 Table” should be revised to “Other extracts used in the screening are shown in S1 Table.”

**Do you want your identity to be public for this peer review?** For information about this choice, including consent withdrawal, please see our Privacy Policy

Reviewer #2: No

---

## [Author Response · Author response to Decision Letter 2]

7 May 2025

We wish to express our appreciation to the editor for their insightful comments on our paper. The comments have helped us significantly improve the paper.

1. The authors agreed that what is reported is additive rather than synergistic. However, the title of Figure 1 still maintains synergistic. I will suggest that all synergistic effects in the manuscript be changed to additive effects since there was no fold decrease in the melanin levels.

Response: We thank the editor for this comment. In accordance with the editor’s comment, we have changed this text to:

Fig 1. SGE and AG additively reduce melanin production.

2. In the materials and methods section, “Other extracts using screening were shown in S1 Table” should be revised to “Other extracts used in the screening are shown in S1 Table.”

Response: Thank you for pointing this out. This error has been corrected in accordance with the editor's comment.

Thank you again for your comments on our paper. We trust that the revised manuscript is suitable for publication.

---

## [Editor Report · Decision Letter 2]

12 May 2025

Sage extract and ascorbic acid derivative inhibit melanogenesis via downregulating keratinocyte-derived GM-CSF

PONE-D-25-07108R2

Dear Dr. Moriyama,

We’re pleased to inform you that your manuscript has been judged scientifically suitable for publication and will be formally accepted for publication once it meets all outstanding technical requirements.

Kind regards,

Anju George

Academic Editor

PLOS ONE

Additional Editor Comments (optional):

The reviewers' comments have been satisfactorily addressed. 
---

## [Editor Report · Acceptance letter]

PONE-D-25-07108R2

PLOS ONE

Dear Dr. Moriyama,

I'm pleased to inform you that your manuscript has been deemed suitable for publication in PLOS ONE. Congratulations! Your manuscript is now being handed over to our production team.

Kind regards,

on behalf of

Dr. Anju George

Academic Editor

PLOS ONE